

# Snapshot fecal survey of domestic animals in rural Ghana for *Mycobacterium ulcerans*

Nicholas J. Tobias[1], Nana Ama Ammisah[2], Evans K. Ahortor[2,3], John R. Wallace[4], Anthony Ablordey[2] and Timothy P. Stinear[1]

[1] Department of Microbiology and Immunology, University of Melbourne at the Peter Doherty Institute for Infection and Immunity, Melbourne, Australia
[2] Department of Bacteriology, Noguchi Memorial Institute for Medical Research, University of Ghana, Legon, Ghana
[3] School of Pharmacy & Pharmaceutical Science, Cardiff University, Cardiff, United Kingdom
[4] Department of Biology, Millersville University of Pennsylvania, Millersville, PA, United States

## ABSTRACT

Identifying the source reservoirs of *Mycobacterium ulcerans* is key to understanding the mode of transmission of this pathogen and controlling the spread of Buruli ulcer (BU). In Australia, the native possum can harbor *M. ulcerans* in its gastrointestinal tract and shed high concentrations of the bacteria in its feces. To date, an analogous animal reservoir in Africa has not been identified. Here we tested the hypothesis that common domestic animals in BU endemic villages of Ghana are reservoir species analogous to the Australian possum. Using linear-transects at 10-meter intervals, we performed systematic fecal surveys across four BU endemic villages and one non-endemic village in the Asante Akim North District of Ghana. One hundred and eighty fecal specimens from a single survey event were collected and analyzed by qPCR for the *M. ulcerans* diagnostic DNA targets IS*2404* and KR-B. Positive and negative controls performed as expected but all 180 test samples were negative. This structured snapshot survey suggests that common domestic animals living in and around humans do not shed *M. ulcerans* in their feces. We conclude that, unlike the Australian native possum, domestic animals in rural Ghana are unlikely to be major reservoirs of *M. ulcerans*.

Corresponding author
Timothy P. Stinear,
tstinear@unimelb.edu.au

## INTRODUCTION

Buruli ulcer (BU) is a neglected tropical disease caused by infection of subcutaneous tissue with *Mycobacterium ulcerans*. Association with swamps and slow-flowing fresh water is a recognized risk factor for contracting BU (*Aiga et al., 2004*; *Pouillot et al., 2007*; *Raghunathan et al., 2005*). Ecological assessments of aquatic environments in Cameroon over 12 months detected the presence of *M. ulcerans* DNA across diverse aquatic invertebrate taxa (*Garchitorena et al., 2014*).

However, field surveys for the pathogen conducted by many groups in different countries do not equally support lentic environments as major reservoirs of *M. ulcerans*

(*Amissah et al., 2014*; *Eddyani et al., 2008*; *Eddyani et al., 2004*; *Gryseels et al., 2012*; *Lavender et al., 2008*; *Vandelannoote et al., 2010*). In BU endemic areas of south-eastern Australia, *M. ulcerans* infects native possums, causing ulcerative disease but also asymptomatic infection with colonization of the gastrointestinal tract (*Fyfe et al., 2010*). In Australian BU endemic areas, >30% of possums can be infected with *M. ulcerans* and they can shed up to $10^6$ bacteria per gram of fecal material, implicating them as major reservoir species (*Carson et al., 2014*; *Fyfe et al., 2010*). Furthermore, the presence of *M. ulcerans* in possum fecal material can predict the occurrence of human BU cases (*Carson et al., 2014*).

We wondered, given there are no possums in Africa, if other animals are acting as *M. ulcerans* source, sink or spill-over reservoirs and shedding contaminated faecal material into their surrounding environments, including local waterways. However, surveys of potential wild, small mammal hosts for *M. ulcerans* have thus far been negative (*Durnez et al., 2010*; *Narh et al., 2015*). Here, we addressed the hypothesis that in African BU endemic villages, common domestic animals such as chickens, goats, sheep, pigs and dogs are potential reservoirs of *M. ulcerans.* We conducted field research in the Asante Akim North (AAN) district, within the Ashanti region of Ghana. Agogo (population 30,000) is the principal town in the AAN district. Five of the communities in the district (Ananekrom, Serebouso, Nshyieso, Bebuso and Dukusen) report the highest burden of BU in Ghana, with 60–75 out of the 120 new laboratory-confirmed cases from Agogo and surrounding districts cases reported annually in these villages (*Ablordey et al., 2015*).

## MATERIALS AND METHODS

### Environmental sampling

One hundred and eighty samples were collected during a single field trip between September 11th and 13th 2013 from four endemic villages (Ananekrom, Afrisere, Dukusen and Nhyieso) and one non-endemic village (Pataban) (Fig. 1). Villages were selected based upon epidemiological data from the Agogo Presbyterian Hospital records. Sampling of sites was performed in a systematic manner, using 10-meter intervals along straight line transects, attempting to traverse the geographic center of any given village (Fig. 1). Chickens (avian), goats and sheep (ovine) are the dominant domestic animal species throughout these villages, so the majority of fecal samples originated from these animals but fecal material was also collected from other sources where available, including dogs and lizards (Table 1). No assumptions were made as to which type of fecal specimen might be *M. ulcerans* positive, with samples collected based on their presence at each 10-meter sample point along the linear transect. GPS coordinates for all sample sites were logged (Table S1) and mapped using Google Earth (v7.1.2.2041). Presumptive source attribution of fecal material was based on physical stool characteristics and appearance.

### DNA extraction from fecal material

DNA was extracted from a maximum of 200 mg of fecal material using the MoBio PowerSoil DNA extraction kit following the manufacturer's recommendations. To guard against false positive and negative results, several types of controls were employed. Extraction contamination was monitored by including pre-tested negative *Pseudocheirus peregrinus*

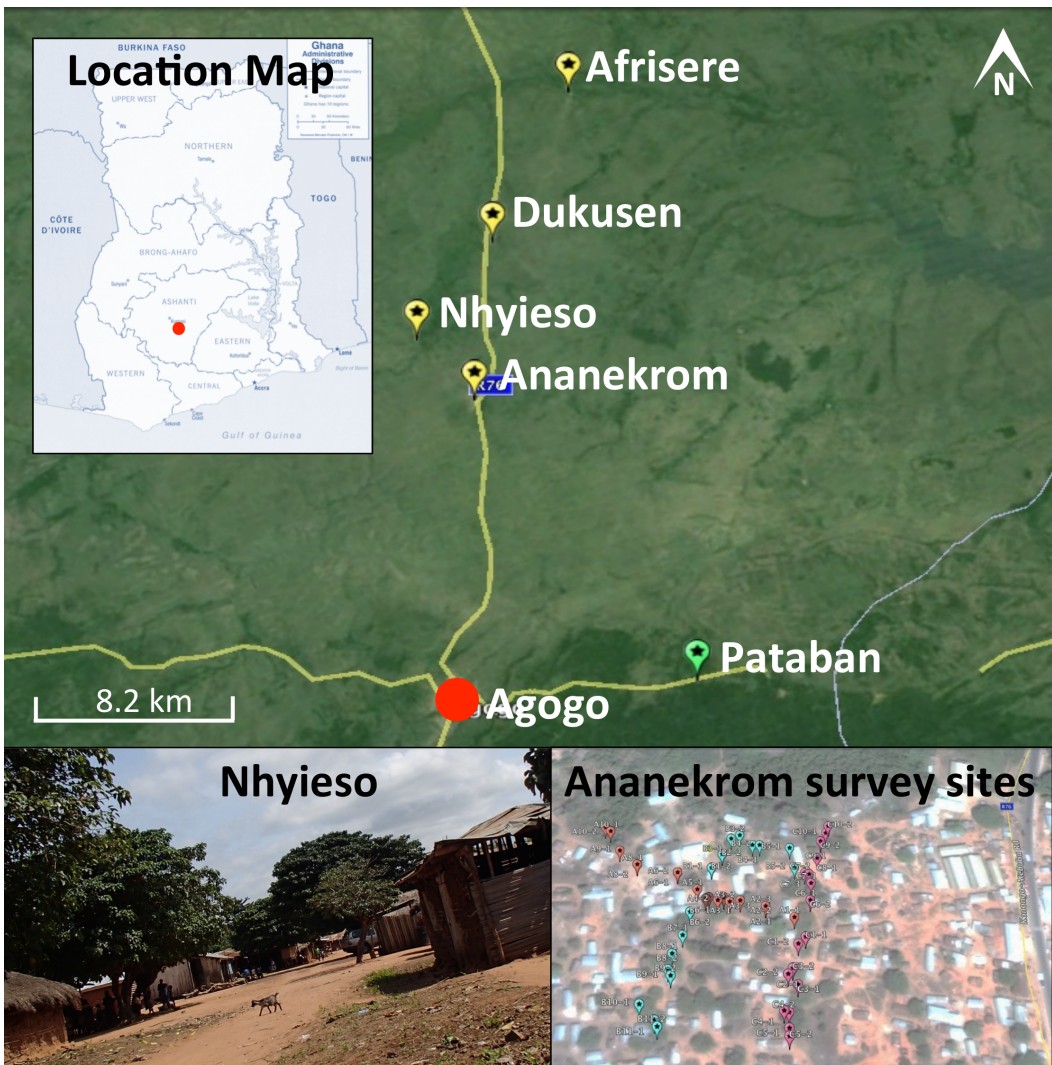

**Figure 1** **Map of the region under investigation across the AAN district within the Ashanti region of Ghana.** Indicated are the Buruli ulcer endemic villages (yellow flags) and non-endemic (green flags) in the AAN district. Shown is an example of the region samples (village of Nhyieso) and three sampling transects in Ananekrom, with sample sites represented by colored flags (red, blue and pink position markers).

(common ringtail possum) fecal material, at a frequency of one control for every 10 samples. A positive extraction control from pre-tested positive *P. peregrinus* feces was also included. Laboratory analysts were blinded to the identity of all control and test samples.

## Quantitative PCR (qPCR)

A 2 μl volume of template DNA was used for Taqman qPCR as previously described, targeting IS*2404* and KR-B (*Fyfe et al., 2007*). A positive sample was defined as positive for both targets with cycle thresholds (Ct) of less than 34 and 36 for IS*2404* and KR-B, respectively, corresponding to a detection limit of approximately 500 *M. ulcerans* genome equivalents, based on reference to a standard curve (data not shown) (*Fyfe et al., 2007*). Samples with an IS*2404* Ct under 34 were tested a second time for

**Table 1  Summary of IS*2404* and KR-B qPCR results separated by fecal sample type.**

| Sample source | IS*2404* positive | KR positive |
| --- | --- | --- |
| Ovine | 2/86 | 0/86 |
| Porcine | 0/1 | 0/1 |
| Avian | 1/69 | 0/69 |
| Reptile | 1/16 | 0/16 |
| Canine | 0/6 | 0/6 |
| Human | 0/1 | 0/1 |
| Unknown | 0/1 | 0/1 |
| Positive control | 1/1 | 1/1 |
| Negative controls | 0/20 | 0/20 |
| Total | 5/201 | 1/201 |

confirmation. In agreement with other research groups using strict acceptance criteria, only those samples positive both times and for both targets were considered *M. ulcerans* positive (*Garchitorena et al., 2014*). The assay includes an internal positive control to monitor potential assay inhibition (*Fyfe et al., 2007*).

## RESULTS AND DISCUSSION

The 180 fecal specimens were collected from a single sampling event across five villages. The sampling was conducted in early September; a time of year associated in the nearby endemic country of Cameroon with increased BU activity or increased likelihood of *M. ulcerans* environmental detection (*Carolan et al., 2014a*; *Garchitorena et al., 2014*; *Landier et al., 2015*). None of the 180 test samples were *M. ulcerans* positive according to our criteria (Table 1, Table S1). Four samples were positive for IS*2404* only. Two of these samples originated from Pataban (a non-endemic village), while the other two were from Nhyieso (endemic village), with all four samples returning Ct values near the 34-cycle threshold cutoff (Table S1). Only the extraction positive control samples tested positive for both IS*2404* and KR-B, below the maximum Ct thresholds (Table 1, Table S1). The strict criteria we employed for a sample to be considered positive ensured that any qPCR positive result was due to the presence of *M. ulcerans* in a given fecal specimen (see 'Materials and Methods').

Faecal samples tested were predominantly of ovine and avian origin because they were the most abundant (47% and 38% of all samples respectively), with samples from other species collected where they occurred within a sampling zone. With the exception of the domestic porcine sample, fecal specimens were obtained from animals free to roam during the day in the immediate environs of villages, including access to local water sources. The sample transects were aligned through the center of villages in an effort to standardize our sampling approach between villages. We conclude from this survey that *M. ulcerans* was not present in the fecal material shed by domestic animals and other peridomestic small animals living in and around the BU endemic villages surveyed in this study. It seems that domestic animals in these villages are unlikely to be reservoirs

of *M. ulcerans* and thus involved in transmission in an analogous manner to that seen in Australian native possums, recalling that the same sampling effort conducted in south east Australia would return 30–40% possum faecal sample positivity rate for *M. ulcerans* DNA (*Carson et al., 2014*; *Fyfe et al., 2010*). A survey for *M. ulcerans* in human fecal specimens from Ghana also provided negative results (*Sarfo et al., 2011*). These observations lead us to summarize that the presence of *M. ulcerans* in the gastrointestinal tract of the native possum of south-east Australia is probably a specific local phenomenon, perhaps representing a particular susceptibility of this animal to mycobacterial infection.

In African BU endemic regions, it has not been possible to predict when and where disease outbreaks will occur and this combined with the long incubation time of the pathogen make the search for possible environmental reservoirs challenging. Nevertheless, independent ecological modeling in Benin and Cameroon, shows that specific land cover and topographical features of a watershed can be used to predict the distribution of *M. ulcerans* (*Carolan et al., 2014b*; *Wagner et al., 2008*), and these data could guide source-tracking efforts. In addition, seasonal variations in the occurrence of the pathogen and the disease have been reported (*Garchitorena et al., 2014*; *Landier et al., 2015*; *Landier et al., 2014*). Our study was only performed at a single time point, so longitudinal fecal sampling of domestic and perhaps feral animals such as the greater cane rat (*Thryonomys swinderianus*) and other large rodents in these regions could be performed to examine the possibility of seasonal pathogen occurrence. With respect to rodents, a survey from Ivory Coast reported *M. ulcerans* DNA in two different stool specimens from *T. swinderianus*, although the predicted concentration of *M. ulcerans* in these samples was very low (*Tian et al., 2016*).

Careful environmental sampling surveys shows *M. ulcerans* DNA present in aquatic environments, but the frequency of positive sample occurrence and the concentration of bacteria in a positive sample are always low. For example, a recently reported environmental survey from Benin conducted over 6 months showed only (28/322 pooled samples from 3,377 aquatic insects were *M. ulcerans* positive) (*Zogo et al., 2015*). Similar low-frequency positivity was observed for aquatic plants and PCR-screening of 942 pools of DNA from flying insects (including 4,322 mosquitoes) tested negative for *M. ulcerans* DNA by PCR (*Zogo et al., 2015*). A survey of 460 water, soil, plant and faecal specimens in Ivory Coast, also using IS2404 qPCR, arrived at similar conclusions with only 2% of samples positive and with only low concentrations of *M. ulcerans* (*Tian et al., 2016*). The few exceptions to this observation serve to reinforce how difficult it is to detect evidence of *M. ulcerans* in aquatic environments (*Bratschi et al., 2014*). Therefore the low burden of the pathogen in such samples raises the question that perhaps lentic environments do not harbour the predicted niche reservoir of this pathogen (*Stinear et al., 2007*). It is the case that—at least in African BU endemic countries—humans infected with *M. ulcerans* are major reservoirs because of the large bacterial burden present in lesions. Humans may not be primary source reservoirs but BU transmission could nonetheless occur indirectly via bacterial contamination of community water sources during bathing of an infected person. Transmission to a naïve individual might then occur via a penetrating injury across a contaminated skin surface (*Meyers et al., 1974*) or via a biting aquatic insect transiently contaminated with *M. ulcerans*.

This hypothesis is entirely consistent with low-level *M. ulcerans* presence in certain aquatic environments and it predicts that interventions designed to reduce the burden of human BU will break transmission chains within a region. Active case finding and prompt antibiotic treatment will decrease *M. ulcerans* shed from human lesions and thus decrease the bacterial burden shed into surrounding environments, whether those environments be aquatic or terrestrial, animal or plant, and thus reduce disease transmission.

Transmission pathways of BU in Africa appear complex and unfortunately remain obscure. The collective efforts of a handful of research teams around the world will need to continue if we are to edge closer to a useful understanding of the means by which this disease is spreading.

## ACKNOWLEDGEMENTS

We are grateful to Janet Fyfe for expert technical assistance and Mohammed Abass for field work support.

### Funding

This work was funded in part by the Stop Buruli Consortium, an initiative of the UBS-Optimus Foundation. The funders had no role in study design, data collection and analysis, decision to publish, or preparation of the manuscript.

### Grant Disclosures

The following grant information was disclosed by the authors:
UBS-Optimus Foundation.

### Competing Interests

The authors declare there are no competing interests.

### Author Contributions

- Nicholas J. Tobias conceived and designed the experiments, performed the experiments, analyzed the data, wrote the paper, prepared figures and/or tables, reviewed drafts of the paper.
- Nana Ama Ammisah and Evans K. Ahortor performed the experiments, reviewed drafts of the paper.
- John R. Wallace conceived and designed the experiments, analyzed the data, reviewed drafts of the paper.
- Anthony Ablordey conceived and designed the experiments, contributed reagents/ materials/analysis tools, reviewed drafts of the paper.
- Timothy P. Stinear conceived and designed the experiments, performed the experiments, analyzed the data, contributed reagents/materials/analysis tools, wrote the paper, prepared figures and/or tables, reviewed drafts of the paper.

## Data Availability

The research in this article did not generate any raw data.

## Supplemental Information

Supplemental information for this article can be found online at http://dx.doi.org/10.7717/peerj.2065#supplemental-information.

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
