# Peer review of "Snapshot fecal survey of domestic animals in rural Ghana for Mycobacterium ulcerans"

_PeerJ, doi:10.7717/peerj.2065_

## Round 0.1 · original submission · Major Revisions

· Academic Editor

Major Revisions

Thank you for your submission. The reviewers have both come back with some criticisms and concerns that need to be addressed upon your revision.

Reviewer 1 ·

Basic reporting

No comments

Experimental design

No comments

Validity of the findings

No comments

Additional comments

Elucidation of M. ulcerans mode(s) of transmission and environmental reservoir(s) is essential for the development of strategies to control and prevent Buruli ulcer outbreaks. The prevailing belief is that the environmental reservoir for M. ulcerans is related to aquatic ecosystems. However, previous reports have shown that M. ulcerans is able not only to infect small mammals in Australia, but also survive and replicate within their gastrointestinal tracts, raising the possibility that mammals play a role in the ecology of M. ulcerans.
In order to develop a more comprehensive understanding of M. ulcerans life cycle and its mode of transmission in an African setting, the authors investigated whether domestic terrestrial mammals could act as a reservoir of M. ulcerans. By using fecal surveys across four Buruli ulcer endemic villages, the authors conclude that domestic animals in rural Ghana are unlikely to be major reservoirs of M. ulcerans.
The manuscript is very well written and the data is clearly exposed. Importantly, sound criteria were employed to avoid false-positives. Nevertheless, there are some aspects that should be clarified, according to the specific comments below:

1- Although I agree with the conclusion that domestic animals are unlikely to be involved in the direct transmission of M. ulcerans, is it correct to assume that they are not major reservoirs solely based on a fecal survey? Could it be possible that these domestic animals are asymptomatically/latently infected and act as a reservoir for haematophagous vectors?

2- What is the prevalence of Buruli ulcer in the endemic regions studied? Is the number of samples collected high enough to detect a positive sample, taking into consideration the prevalence of Buruli ulcer among humans in this endemic region?

3- What are the feeding/grazing/drinking habits of the domestic animals tested? Given that M. ulcerans has been predominantly linked to aquatic ecosystems, it would be important to know whether these animals have been in close proximity to water sources and possibly exposed to M. ulcerans.

4- Do the authors envisage different results if sample collecting were to occur during the dry vs. wet season? Have previous studies linked Buruli ulcer reports with seasonality?

Reviewer 2 ·

Basic reporting

The article is well written and the introduction is sufficient for those familiar with the field, data has been made available and figures are legible.

Experimental design

The paper is within the scope of the journal, identifies a clear research question and methods are reproducible by the reader.

However, I have concerns about the validity of the environmental sampling, as discussed below.

Validity of the findings

The data are robust in terms of the output of the qPCR analysis, insofar as I can tell, however statistically there is a concern with sampling bias. This means the conclusions drawn from the data are overreaching and need to instead be limited to those conclusions that can be supported by the results.

Additional comments

This is the first time I’ve reviewed an article for PeerJ, I like the idea of the journal and find it to be quite a novel approach to publishing. I’ve gone through the editorial criteria the aims and scope of the journal, and the paper fits these well, except for a concern about experimental design as discussed below.
Generally, I like the paper and commend the authors on their work. However, I have two concerns that have to be addressed before the paper is ready for publication. The first is a limitation in the sampling protocol that has not been discussed. Second, given that further field work is unlikely to be possible to rectify the sampling protocol, the conclusions that have been drawn are overreaching. Discussing the impact of the sampling strategy in the limitations section and presenting the conclusions in a different way should help to address this issue. Finally, Buruli ulcer is a neglected disease and I commend the authors for their efforts in eradicating the problem, including what must have been truly gruelling field work!

1) the environmental sampling
The DNA extraction and qPCR protocols seem robust and sound. However, I have concerns with the environmental sampling. The authors collected 200 samples (which is an impressively large number), however I have a concern with the position of the transects.

M. ulcerans is generally aquatic, and associated with wet, dark, stagnant environments that have been disturbed. Your sample sites don’t reflect this. First of all, I can imagine that a stool sample laying in the street of an urban environment for an unknown amount of time would rapidly dry out, in baking heat with high levels of UV and oxygen, not to mention the complex of fungi and bacteria that will rapidly be degrading the sample. This is not the kind of environment that M. ulcerans or its DNA can survive in. For example, if stool samples were to be collected from peri-aquatic, shaded regions, would the results be different?

Secondly, these animals are mobile. Goats and dogs will visit water sites to drink and bathe, where they may, perhaps, acquire the bacterium. Are they drinking from infected water sites or are they drinking from clean sources (eg wells or rainwater vats)? From figure 1 it appears the samples are all in the same areas, even along the same streets. This may be quite appropriate for sampling possum scat in Australia (as previous work by these authors has shown), however compare this to the environmental sampling by Marion, Garchitorrena or Morris – which standardly were several km apart. Consider also Carolan 2014 which discussed the importance of watersheds as a limit to the population distribution of M. ulcerans. I am afraid that the 200 samples are probably coming from the same population of domestic animals – if they are all exposed to the same aquatic source, which may be different to humans, are they likely to be infected in the first place?

The type of sample you are collecting, and the locations you are collecting from, may be biasing you towards detecting negatives. If the samples were to be collected on a larger scale they may reveal a different pattern. Strictly speaking this does not at all invalidate your results – they are true for the scale the study was done on (I believe your DNA extraction and qPCR protocol, so I agree that the samples are negative) however there is then a mismatch with your conclusion, as below.

2) the conclusions
You are right that the limitation in time may be an important factor, however I believe most people would concede that such temporal studies are practically very difficult. This does not invalidate the findings, but simply means we cannot extrapolate in time, just as we cannot extrapolate across space or scale. Your conclusion that M. ulcerans is not present in the fecal material of domestic animals in BU endemic regions at this time point is still valid.

I am afraid I must disagree entirely with the rest of your conclusion, and feel that a lot more evidence is needed to justify it (I suspect this will land me the position of reviewer #2!)

Your statement on line 134 that “humans infected with M. ulcerans represent the major source reservoir for M. ulcerans” is in contradiction with the literature. Insofar as I am aware, neither human to human nor human to environment transmission (at significant rates) has been observed. If it has please cite appropriate literature. Humans may contain large numbers of M. ulcerans however they can be deadend hosts if the bacterium is not able to re-infect the environment to a significant degree. As a loose analogy: the highest concentrations of krill in the ocean are in the mouths of blue whales – that doesn't make the mouth of the whale the ecological niche of the krill!

The work of the Guegan group, the Marsollier group and the Benbow group all point to the idea that the major source of M. ulcerans in West Africa is environmental, not human.
The recent spate of papers by Garchitorrena et al, Morris et al, Garcia-Pena, Marion and Carolan et al in the last few years demonstrating the very wide distribution of M. ulcerans in environmental networks of aquatic insects and the broader aquatic community discuss this in more detail.

The supposed role of mosquitoes in transmission needs to be justified. In the same way that we would not be justified in extrapolating from Australia to West Africa by saying that mammals are the primary host, we are also unjustified in extrapolating that mosquitoes are the mechanism of transmission. That is not to say they are not related to transmission in Australia, I look forward to the work of Wallace et al, but in a different context, with different hosts, different climate conditions, it still has to be demonstrated that they are implicated in transmission in West Africa.

The prediction that reduction of BU in humans will break the infection chain is also unjustified. For example, if in Australia the possum is the primary host, and supposing mosquitos acquire the bacterium and transmit it to humans, reduction of human BU may not lead to reduction of possum BU, which means the (presumably mostly unidirectional) transmission chain will not be broken.

You in fact contradict yourself in the last sentence. You say that prompt case detection ought to result in a decrease in the environmental burden of M. ulcerans – M. ulcerans distribution is known to be greater than the distribution of Buruli ulcer in West Africa. There is no evidence that eliminating the disease from humans will remove the bacteria from the environment. By analogy, prompt treatment of tetanus does not reduce the prevalence of Clostridium tetani in the environment.


Final summary
On rereading my review I feel I have been somewhat harsh – I would like to reassure the authors that I generally do feel this is suitable for publication and I trust the integrity of their DNA protocol, but I feel their environmental sampling needs to be addressed as a limitation, and this limitation means their conclusion overstates the possibility of vector transmission.

---

## Round 0.2 · accepted · Accept

· Academic Editor

Accept

The re-review has now come back and my decision is to accept.

Reviewer 2 ·

Basic reporting

No Comments

Experimental design

No Comments

Validity of the findings

No Comments

Additional comments

1st response,
I bow to the authors experience with DNA degradation (which is not my own area of expertise). Certainly it seems that if the DNA is able to survive long enough the sampling protocol is more robust.

2nd response,
I agree, and given your response to reviewer 1’s point that the rate of BU is high, it seems at least probable that there is some about of bacterium in the water. As a suggestion for future studies, if it is practical of course, perhaps complementing the faecal sampling with aquatic sampling might be informative (simply as a suggestion!)

3rd & 4th responses,
Apologies, I did not explain the point clearly.
Suppose you have a population of goats, whose movement and habits expose them at a certain rate to MU. Assuming all goats in the population are equally likely to be exposed, then we need repeated samples of the population to tell the level of infection in the population; if I understand this properly then the transect is the repeated samples (different animals, as you say) – you then need to compare between enough populations – which is 4 villages.
It’s actually a bit of an unfair criticism on my part – the ever returned to “more samples could show a different pattern”.
However, I take your point that they draw from different water sources to be reassuring. As a suggestion (again, simply a constructive suggestion) for future designs, a power analysis first may be helpful.

5th and 6th response
Your reply and the addition to the text in the paper is quite exciting, and I have to say I disagree with you, however you make your case well, so I’ll bite my tongue.
As a technical point, I think that the observations that land cover and climate can drive BU in humans is a point against your argument, however that is certainly not within the scope of this paper.

7th response
Apologies, I must have misunderstood, I took this from your sentence “BU transmission may then occur via fomite contamination from an infected human to a naïve individual, perhaps via blood-feeding insects”

8th response
I agree that the PCR beyond the human population is low, I guess this comes down to the idea that the impact of the bacterium is directly though numbers rather than locations (my suspicion is that a low concentration of bacteria in one situation eg an insect vector has a higher chance of infecting humans than a high concentration in a different situation, eg within an ulcer, maybe)
Your point that treating the cases promptly may reduce environmental load is certainly worth following, and even if I am sceptical of the role of humans, everyone would agree that prompt treatment of the disease is certainly a good thing anyway.